# ACTIVE PARTITIONING:
# INVERTING THE PARADIGM OF ACTIVE LEARNING

## ABSTRACT

Datasets often incorporate various functional patterns related to different aspects or regimes, which are typically not equally present throughout the dataset. We propose a novel, general-purpose partitioning algorithm that utilizes competition between models to detect and separate these functional patterns. This competition is induced by multiple models iteratively submitting their predictions for the dataset, with the best prediction for each data point being rewarded with training on that data point. This reward mechanism amplifies each model's strengths and encourages specialization in different patterns. The specializations can then be translated into a partitioning scheme. The amplification of each model's strengths inverts the active learning paradigm: while active learning typically focuses the training of models on their weaknesses to minimize the number of required training data points, our concept reinforces the strengths of each model, thus specializing them. We validate our concept – called active partitioning – with various datasets with clearly distinct functional patterns, such as mechanical stress and strain data in a porous structure. The active partitioning algorithm produces valuable insights into the datasets' structure, which can serve various further applications. As a demonstration of one exemplary usage, we set up modular models consisting of multiple expert models, each learning a single partition, and compare their performance on more than twenty popular regression problems with single models learning all partitions simultaneously. Our results show significant improvements, with up to 54% loss reduction, confirming our partitioning algorithm's utility.

## 1 INTRODUCTION

Datasets can include multiple sections that adhere to distinct regimes. For instance, in stress-strain tests of materials, the initial phase exhibits elastic behavior, which is reversible. However, if the material is stretched further, it enters a phase of plastic behavior, resulting in permanent changes. Similarly, self-driving cars face unique challenges when navigating construction zones, which may be specific to certain regions of the parameter space, just as the challenges on highways or country roads. This mixture of different functional patterns within datasets affects how difficult they are for models to learn. Typically, the more diverse the functional patterns within a dataset, the more challenging it is for a model to achieve high accuracy. We present a novel partitioning algorithm aiming to detect such different functional patterns and separate them from one another whenever possible.

The development of algorithms for partitioning datasets has a long history. MacQueen introduced the renowned k-means algorithm in 1967 (Macqueen, 1967). However, most approaches define an arbitrary similarity measure for grouping data points. In k-means, spatial proximity is interpreted as the similarity of data points. In contrast, we allow those models that are supposed to learn the dataset to determine which data points they can coherently learn. We believe that for effective dataset partitioning, it is crucial to consider the models themselves. As Jacobs (1990) stated, "the optimal allocation of experts to subtasks depends not only on the nature of the task but also on that of the learner".

Our new algorithm is based on the competition between multiple models. These models are iteratively trained using the data points for which they have made the best predictions, thereby emphasizing each model's strengths and inducing model specialization. The models being trained specifically

on their strengths, rather than their weaknesses, inverts the traditional active learning strategy, leading us to term this approach active partitioning. The resulting data point distribution to different models is translated into partitions representing different regimes or patterns. Technically, the outcome of this algorithm is the boundaries between these partitions, stored in a support vector machine (SVM).

There are several ways to utilize the resulting partitioning. One notable application is to learn each partition with a separate model and then combine these expert models into a modular model, allowing each expert model to focus on a specific pattern rather than handling all patterns simultaneously. Our experiments demonstrated that such a modular model, based on the results of the partitioning algorithm, significantly outperformed a single model on several exemplary datasets.

## 2 RELATED WORK

### 2.1 PARTITIONING

Extensive literature surveys on clustering, including partitioning as a special form, can be found in Jain (2010), Du (2010), Aggarwal & Reddy (2013), and Ezugwu et al. (2021). According to Jain (2010), clustering is about identifying groups within datasets such that "the similarities between objects in the same group are high while the similarities between objects in different groups are low." There are four major categories of clustering algorithms: hierarchical algorithms, partitional algorithms, density-based algorithms, and heuristic algorithms. This paper focuses on partitional algorithms, which dynamically assign data points to clusters in either a hard or soft manner. In hard partitional clustering, each data point is assigned to exactly one cluster, whereas in soft partitional clustering, each data point can belong to multiple clusters.

K-means is the most well-known partitional clustering algorithm. In each iteration, each data point is assigned to the nearest centroid, after which each centroid takes over the main position of all its data points (Macqueen, 1967). To address weaknesses such as the need to specify the number of clusters in advance or the convergence to a local minimum, the algorithm can be run multiple times with different numbers of clusters and random initializations. A prominent extension of k-means is fuzzy c-means, which allows for soft partitional clustering (Dunn, 1974). A recent extension is game-based k-means, which increases competition between centroids for samples (Rezaee et al., 2021).

In the 1990s, the kohonen network and its specialization, the self-organizing map, were developed. Both consist of two neural network layers: an input layer and an output layer, known as the kohonen layer. The prototypes competing for data points are the neurons in the kohonen layer (Kohonen, 1990). Most approaches that utilize competition for partitioning have entities within one model compete, such as the centroids in k-means or the last-layer neurons in the kohonen network. To our knowledge, Müller et al. are the only exception, aiming to segment temporally ordered data by identifying switching dynamics. They defined an error function and an assumption of the error distribution to trigger competition between neural networks for data points (Müller et al., 1995). Chang et al. extended this framework to use support vector machines instead of neural networks (Chang et al., 2004).

A completely different approach for localizing and specializing experts is the iterative splitting of datasets and models, as suggested by Gordon & Crouson (2008). Zhang and Liu combine model splitting and competition in what they call the "one-prototype-take-one-cluster" (OPTOC) paradigm (Zhang & Liu, 2002). New models are created if the accuracy is not yet satisfactory, and these models then compete for data points, which later defines the localization of the experts. Wu et al. adapted this paradigm for clustering gene expression data (Wu et al., 2004).

The novelty of our research lies in the development of a flexible partitioning method through the competition of entire models. To the best of our knowledge, no general-purpose partitioning algorithm has previously employed competition between entire models. Although the simultaneous training of experts and a gate involves model competition, it has not been used to create a partitioning (Jacobs et al., 1991). Müller et al. utilized competition for data segmentation, relying on the switching dynamics of temporally ordered data (Müller et al., 1995). In contrast, our approach is not constrained by any specific origin or order of data.

## 2.2 COMBINING MODELS

To demonstrate the effectiveness of our algorithm, we will compare a modular model based on our partitioning approach with a single model. Therefore, we will also review related work on combining multiple models. Comprehensive overviews of model combination techniques can be found in Sharkey (1996), Masoudnia & Ebrahimpour (2014), and Dong et al. (2020). Multiple models can either be fused to an ensemble, meaning that they are trained at least partially on the same data and that their predictions are combined by a weighted average, or they can form a modular model, meaning that they are trained on different parts of the dataset and for each prediction only one responsible model is selected. An approach representing a compromise between these two poles is the mixture of experts system, which consists of expert models and an additional gating model mediating the experts, all of which are trained simultaneously. The design of the error function that is minimized is crucial for the extent of localization or specialization of the experts and therefore for the quality and generalization of the overall predictions. Typically, neural networks are selected as models in the mixture of experts system (Jacobs et al., 1991)(Avnimelech & Intrator, 1999). An obvious advantage of a mixture of experts system compared to a single model is the significantly increased capacity to learn large or complex datasets. Shazeer et al. recently combined thousands of neural networks into a sparsely-gated mixture of experts system (Shazeer et al., 2017). Since the gating network only activates a few expert networks per sample, they achieved a dramatic increase in model capacity while even decreasing the computational cost compared to state-of-the-art models.

## 3 PRESENTATION OF THE ALGORITHM

The objective of our approach is to detect functional patterns in datasets and separate them in case they appear separable. To achieve this, we propose competition among multiple models. We intentionally refer to models in a general sense, as our approach is not limited by the type of model used. However, for simplicity, one might consider simple feedforward networks as an example. The models compete for data points, which requires them to specialize in certain functional patterns of the dataset. This specialization can be translated into a partitioning of the dataset.

We assume that the input features and the output labels of the dataset are known. However, we assume that both the number of partitions and the location of their boundaries are unknown.

The algorithm operates as follows: for each data point in the dataset, all models submit their predictions. The model whose prediction is closest to the true value wins the data point. As a reward for providing the best prediction, the winning model is allowed to train on this data point for one epoch. Algorithm 1 describes all the steps mentioned. A corresponding flowchart is shown in Figure 1.

Figure 1: flow chart of the partitioning algorithm: each data pointed is assigned to the model that submitted the best prediction. All models are trained with the data points in their partition for one epoch. This process is iterated.

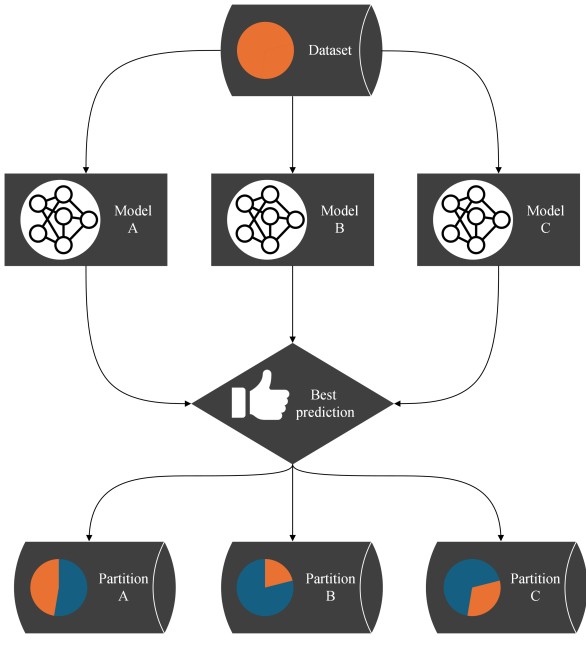

This process — models submitting predictions, ranking the predictions, and training the models on the data points for which they provided the best predictions — is iterated. One iteration we call an epoch of the algorithm. As the models specialize, we expect the assignments of data points to models to stabilize: a specialized expert will usually submit the best predictions for its domain. After

Figure 2: exemplary partitioning. Figure 2a presents the self-designed test dataset, while Figure 2b displays an exemplary partitioning result. Figure 2c illustrates the partitioning process, transitioning from networks with initial random predictions to the orange, red, and green networks each capturing distinct patterns. The process involves adding and removing networks as patterns are identified or networks deemed redundant.

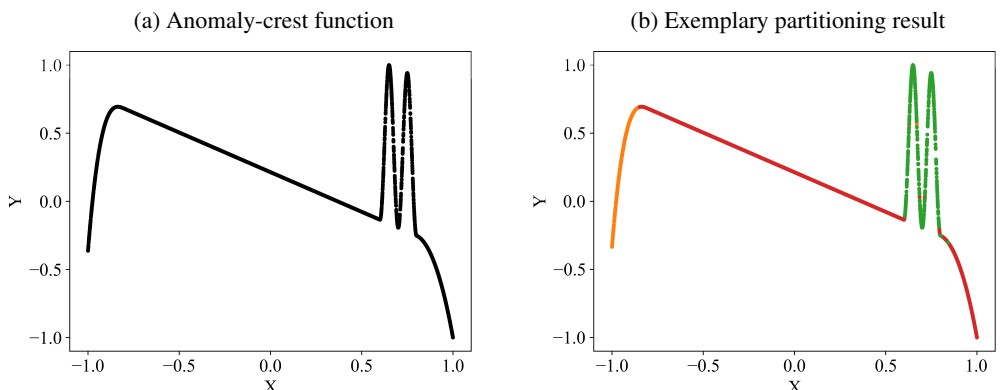

(a) Anomaly-crest function      (b) Exemplary partitioning result

(c) Exemplary partitioning process with the network number evolution over 1000 iterations (epochs)

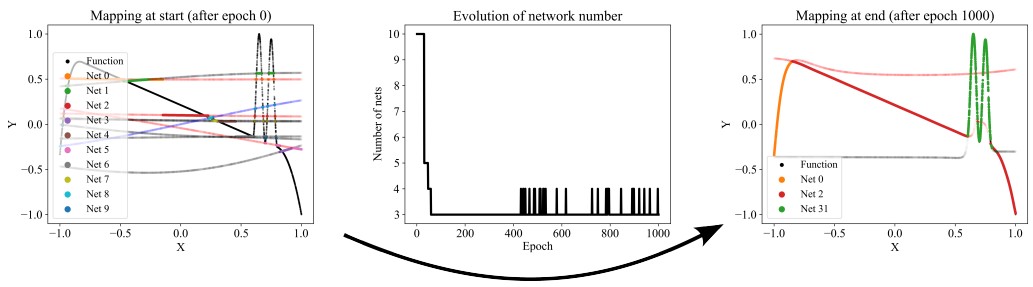

---

**Algorithm 1** Partitioning: best predictions are rewarded with training.

---

**procedure** MAIN
    **for each** *epoch* **do**
        **for each** *model* **do**
            Submit predictions for all data points.
        **end for**
        **for each** *datapoint* **do**
            Rank models according to their predictions.
        **end for**
        **for each** *model* **do**
            Train for one epoch with all won data points.
        **end for**
    **end for**
**end procedure**

---

a predefined number of epochs, the assignments of data points to models are considered final. Each model's won data points translate to a separate partition of the dataset. The hyperplanes between the partitions are stored in an SVM, making the partitioning technically available for other applications. Snapshots of the application of the algorithm to a two-dimensional function that we designed as a test dataset are shown in Figure 2. The transition from random predictions at the beginning to specialized experts at the end is clearly visible. The assignments of data points to the specialized experts are translated into the final partitioning.

Since the number of partitions is usually unknown beforehand, the partitioning algorithm includes an adding and a dropping mechanism to dynamically adapt the number of competing models to the

Figure 3: adding a new network (red network 12) to the competition. Regularly, a new network is trained using the data points with the poorest predictions at that time. If the new network improves the overall loss, it is added to the competition. Here, the red network 12 is the first to capture the sinusoidal pattern.

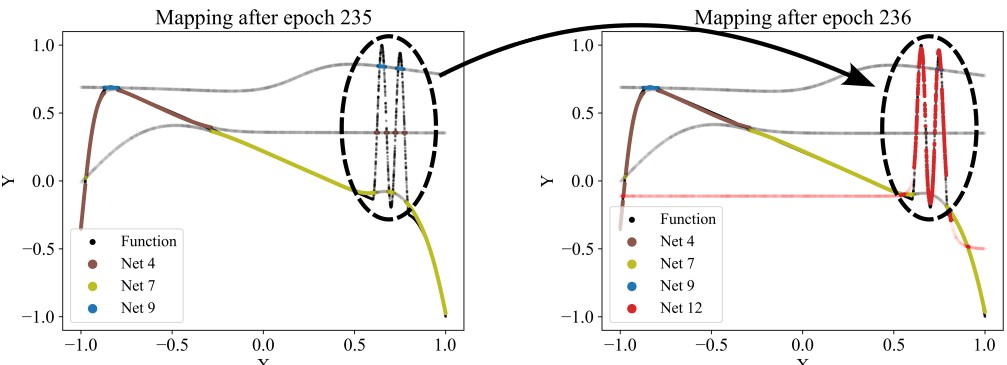

Figure 4: dropping a network (red network 12) from the competition as it appears redundant, failing to capture any patterns uniquely. Regularly, for each model, we check how much the overall loss would increase if the network were removed. If the increase is small, the corresponding network is considered redundant and is discarded. Here, the red network's predictions were too similar to the purple network's predictions.

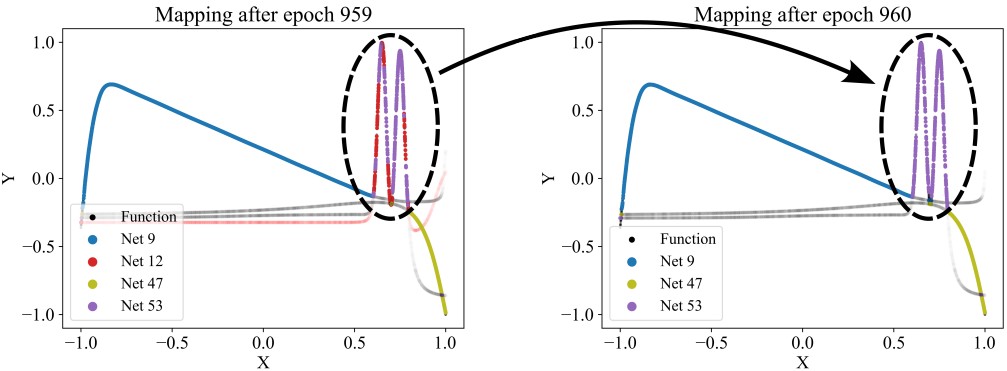

dataset. To evaluate whether a new model should be added to the competition, we regularly identify the data points with the poorest predictions in the dataset and train a new model on these points. The new model is added to the competition in case that improves the overall loss. Figure 3 demonstrates the addition of a model that successfully captures a significant portion of the sinusoidal section of a test function, which had previously been unlearned. For more details, see the pseudo-code of the adding mechanism in Algorithm 2 in the Appendix A.2. Conversely, redundant models that do not uniquely capture their own pattern should be eliminated. Such redundancy is indicated by models not winning any data points or by their predictions significantly overlapping with those of other models. The degree of redundancy is assessed by the increase in overall loss if the model were deleted. This factor is regularly checked, and all highly redundant models are removed. Figure 4 demonstrates the removal of the red model, as it only captures data points similarly well as the purple model. Algorithm 3 in the Appendix A.2 provides the corresponding pseudo-code. The adding and dropping mechanism are designed to balance each other. Figure 2 shows exemplary how the number of competing models is adapted to the dataset from initially ten to finally three. This process involves both adding new models to capture previously unlearned patterns and removing redundant ones.

A significant asset of our partitioning algorithm is its ability to extend to a pattern-adaptive model type, architecture, and hyperparameter search without incurring additional costs. So far, competing models have been considered similar in terms of their type, architecture, and hyperparameter settings. However, all three can be randomly varied among the models, as it is reasonable to assume that different patterns may require, for example, wider neural networks or smaller learning rates. Consequently, the algorithm's output can not only be a partitioning but also an optimal configuration of model type, architecture, and hyperparameters for each partition.

# 4 APPLICATIONS

## 4.1 MODULAR MODEL

Figure 5: flow chart of the modular model: each partition is learned by a separate expert model. For each data point, the SVM as a result of the partitioning algorithm decides which expert to train or to test. This way, the experts are combined to a modular model.

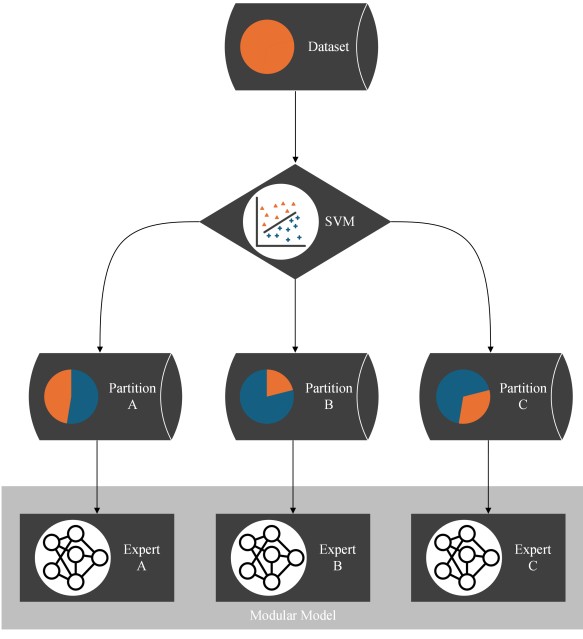

Applying the partitioning algorithm to datasets reveals interesting and valuable insights about the dataset's structure, as illustrated in Figure 2. Additionally, the partitioning can be utilized for various other purposes, such as learning the dataset using a divide-and-conquer approach. Traditionally, the entire dataset is used to train and optimize a single model. However, if the partitioning algorithm detects distinct functional patterns, it may be beneficial to have multiple expert models, each learning only one pattern, instead of pressing all patterns into a single model. Therefore, multiple expert models that each learn one partition are combined into a modular model. The SVM, which incorporates the boundaries between the partitions, serves as a switch between the experts. For each data point, the SVM decides which partition it belongs to and, consequently, which expert model to train or test. The structure of the modular model is illustrated with a flowchart in Figure 5. With this approach, we believe that we can reduce model complexity and increase model accuracy for datasets that are structured by multiple distinct functional patterns with little overlap.

To evaluate this approach, we compared the performance of a single model trained on the entire dataset with that of a modular model comprising multiple expert models. We speak of models in general, as the type of model can be varied. In our experiments, we used feedforward neural networks. To ensure a fair comparison, we allowed the single model to have as many trainable parameters (weights and biases) as the combined total of all experts in the modular model. We conducted a hyperparameter optimization for each expert, searching for the optimal number of layers, neurons per layer, and learning rate within reasonable constraints (see Table 2 in Section A.2). Separately, we performed a hyperparameter optimization for the single model, allowing it to use as many trainable parameters as all the experts combined. Each hyperparameter optimization involved training the model 100 times with randomly varied hyperparameters and selecting the best result. This process ensured that any advantages or disadvantages were not due to unfitting parameters or outliers. To estimate the stability of both approaches, we repeated each run—partitioning the dataset, training the modular model including hyperparameter optimization, and training the single model including hyperparameter optimization—ten times.

Figure 6: datasets to test the partitioning algorithm, illustrated with exemplary partitioning results.

(a) Self-designed wave-climb function with three patterns identified by the algorithm (grey, green, blue).

(b) Porous structure's stress-strain dataset generously provided by Ambekar et al. (2021) with three patterns identified by the algorithm (red, green, orange).

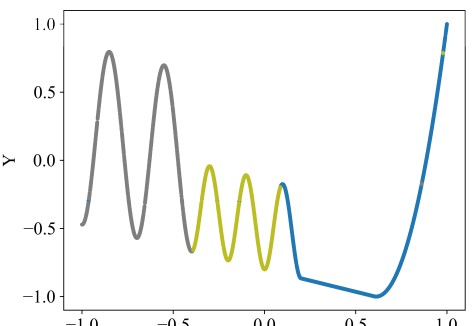
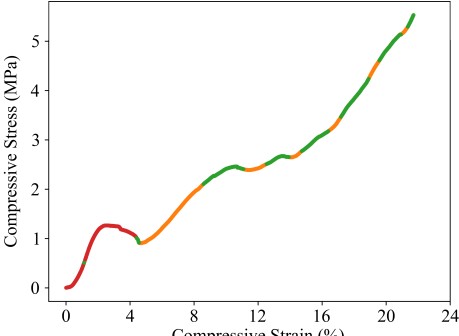

## 4.2 DATASETS

We designed two-dimensional, section-wise defined functions to serve as test datasets for validating the effectiveness of our approach and its implementation. The anomaly-crest function is illustrated in Figure 2a, and the wave-climb function is depicted in Figure 6a. Due to their section-wise definition, these functions exhibit different local functional patterns, akin to several engineering problems. One such example is modeling the stress-strain curves of materials with porous structures. These materials offer an excellent balance between weight and strength, but their stress-strain curves are typically challenging to model due to the presence of diverse functional patterns. An exemplary stress-strain curve for such a material is shown in Figure 6b. The data for this porous structure's stress-strain curve were generously provided by Ambekar et al., who collected them (Ambekar et al., 2021). We have observed a high robustness of our partitioning approach to variations in the models' random initializations. Figures 2 and 6 illustrate typical results.

In addition to the two-dimensional datasets, we evaluated our method using popular higher-dimensional real-world datasets from the UCI Machine Learning Repository (Kelly et al., 2024). Our tests focused exclusively on regression problems, though our approach can be readily extended to classification problems. Acknowledging that our assumption of distinct and separable characteristics may not apply to all datasets, we tested 22 additional datasets to assess the frequency and extent to which the modular model, based on the partitioning algorithm, outperforms a single model (Imran et al., 2020) (Cortez et al., 2009) (Nash et al., 1995) (Palechor & la Hoz Manotas, 2019) (Schlimmer, 1987) (Cortez & Morais, 2008) (Feldmesser, 1987) (Yeh, 2018) (E & Cho, 2020) (Tsanas & Xifara, 2012) (Yeh, 2007) (Tfekci & Kaya, 2014) (Cortez, 2014) (Quinlan, 1993) (Matzka, 2020) (Wolberg et al., 1995) (Fernandes et al., 2015) (Janosi et al., 1988) (Tsanas & Little, 2009) (Tsanas & Little, 2009) (Chen, 2017) (Moro et al., 2016) (Hamidieh, 2018).

## 4.3 RESULTS

Figure 7 presents histograms comparing the test losses of the modular and single models. For this illustration, we selected those 6 out of the 25 datasets for which the modular model achieved a significant advantage over the single model. Each histogram shows the losses from ten runs of both models on each dataset. The x-axis represents the test loss, while the y-axis indicates the number of runs achieving each respective test loss. Higher bars on the left side of the histogram indicate better performance.

The modular model, utilizing the partitioning algorithm, significantly outperforms the single model by orders of magnitude for the two test functions (see Figs. 7a and 7b), demonstrating the concept's validity. For the porous structure's stress-strain data, which inspired the design of the test functions, the modular model achieved a 54% reduction in loss compared to the single model (see Fig. 7c). Additionally, the modular model could significantly outperform the single model for three other real-world datasets: for the energy efficiency dataset, the modular model achieved a 53% improvement

Figure 7: histograms illustrating the test losses of single and modular model for ten runs with each of the six selected datasets. The higher the bars on the left side, the better the performance.

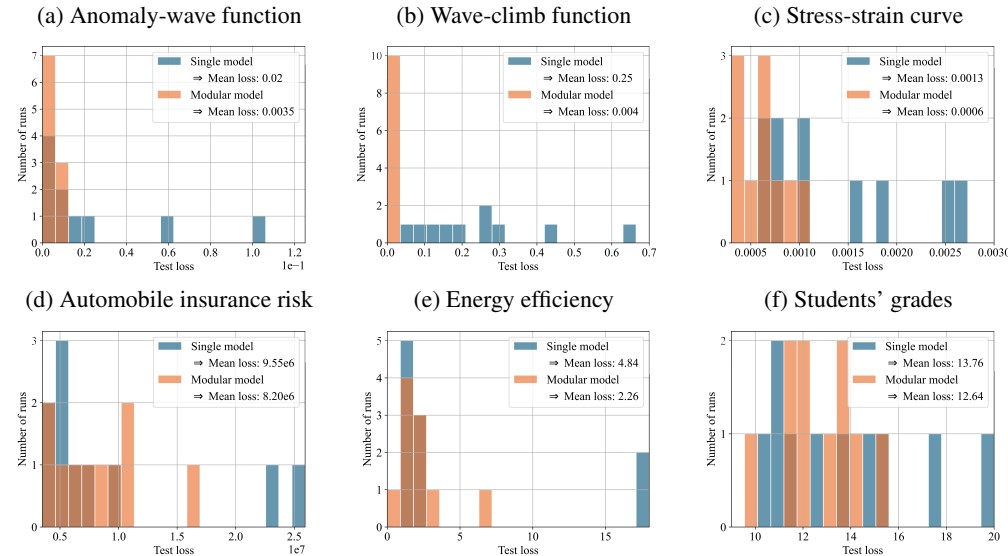

in mean loss over ten runs (see Fig. 7e). For the automobile dataset, the improvement was 14% (see Fig. 7d), and for the dataset on students learning portuguese, the improvement was 8% (see Fig. 7f).

Table 1 provides a brief characterization of each of the six datasets, detailing the number of features, labels, and data points. A more detailed analysis of the modular model's performance compared to the single model can be found in Section A.1.

Table 1: selected dataset characteristics for sparse and non-sparse datasets

|  | Features | Labels | Data points |
|---|---|---|---|
| Anomaly-crest function | 1 | 1 | 10,000 |
| Wave-climb function | 1 | 1 | 10,000 |
| Porous structure's stress-strain curve | 1 | 1 | 4,065 |
| Automobile insurance risk | 59 | 1 | 159 |
| Energy efficiency | 8 | 2 | 768 |
| Students' portuguese grades | 56 | 1 | 649 |

## 5 DISCUSSION

As introduced in Section 3, the partitioning algorithm is based on the competition between multiple models: iteratively, each model is trained on the data points for which it provided the best predictions. This approach inverts typical active learning strategies, which usually focus on training models on their weaknesses to enhance generalization. Instead, we emphasize the strengths of each model to induce specialization. We consider our concept to be the first in a new category of partitioning approaches: active partitioning approaches, which invert the traditional active learning paradigm.

The application of our active partitioning algorithm to the anomaly-crest function (see Fig. 2) demonstrates that the competition between multiple models is generally effective for developing specialized experts and separating different functional patterns. The primary value of this partitioning lies in its ability to detect these distinct patterns and provide insights into the dataset's structure. For the anomaly-crest function, the four identified sections clearly differ in their functional charac-

teristics (see Fig. 2). In the case of the wave-climb function, the algorithm successfully separates the two sinusoidal sections with different frequencies and amplitudes, as well as a final u-shaped section, which seems reasonable (see Fig. 6a). For the porous structure's stress-strain dataset, it is noteworthy that the first hook is identified as a distinct pattern. Subsequently, all sections with concave curvature are captured by the green model, while all sections with convex curvature are captured by the orange model. This partitioning was surprising, but it appears that the models find it easier to learn either concave or convex curvatures exclusively (see Fig. 6b). The models themselves detecting which functional patterns can be learned well coherently was exactly what we were aiming for.

There are several ways to utilize the partitioning, and we found it important to also illustrate a path that leads to measurable improvements by leveraging our partitioning results. In Section 4.1, we introduced modular models that combine multiple experts. Our hypothesis is that for datasets with separable patterns, it may be advantageous to have multiple experts, each focusing on a single pattern, rather than a single model handling all patterns. As expected, the modular model was not superior for all datasets. We believe this is because if a dataset exhibits only one coherent pattern or if multiple patterns highly overlap, it is more beneficial for a single model to access all data points rather than splitting them. However, among the 25 datasets we tested, we identified six datasets that could be learned more precisely with the modular model utilizing our partitioning results. For the porous structure's stress-strain dataset and the energy efficiency dataset, the modular model even achieved a loss reduction of more than 50% (see Fig. 7).

In Section A.1, we describe a detailed analysis of the factors contributing to the performance of the modular model. Our findings reveal a correlation between the number of patterns identified by the partitioning algorithm and the modular model's performance: the more distinct patterns in the dataset, the better the modular model performs relative to the single model. This aligns with our expectation that not all datasets are suitable for our approach. The partitioning algorithm should primarily be applied to datasets that are expected to exhibit predominant patterns with minimal overlap. The clearer the patterns, the more effective the modular model is expected to be.

Additionally, we examined the impact of our pattern-adaptive hyperparameter search, which optimizes the hyperparameter settings for each pattern. We discovered that tailoring the learning rates to each partition enhances the modular model's performance. However, our results indicate that adjusting the numbers of layers and neurons per layer for each pattern does not provide any significant advantage.

Finally, we aimed to verify that the partitioning algorithm identifies substantial patterns rather than merely separating small and challenging snippets. Our results confirm that the more homogeneous the partition proportions, the more successful the modular model tends to be.

There are numerous potential applications, many of which we may not have yet considered. One application we plan to explore is using the partitioning algorithm for active learning. In the context of expensive data points, the following data collection loop could be advantageous: first, collect a batch of data points; then, apply the partitioning algorithm; and finally, train each partition with a separate model, akin to the modular model approach. Instead of immediately combining their predictions, we could assess each expert's performance and adjust the collection of new data points accordingly. Partitions that are more challenging to learn should receive more data points, while easier partitions should receive fewer. This approach could lead to a more efficient use of the data point budget. The process can be repeated iteratively. For instance, with a budget of 500 data points, we could run this process 10 times, each time distributing 50 data points according to the difficulty of the experts in learning their partitions in the last iteration.

## 6 CONCLUSION

In this paper, we introduced a novel active partitioning algorithm. To the best of our knowledge, this algorithm is unique in its use of competition between models to generate a general-purpose partitioning scheme, without constraints on the dataset's origin or order. The partitioning is achieved by having multiple models iteratively submit their predictions for all points in the dataset and being rewarded for the best predictions with training on the corresponding data points. This process induces specialization in the models, which is then translated into a partitioning. Focusing the training on

each model's strengths practically inverts the active learning paradigm of focusing the training on a model's weaknesses, leading us to call our concept an active partitioning algorithm.

We demonstrated that our algorithm is both widely applicable and useful. Its wide applicability was shown by valuable results across datasets of varying dimensionalities, sparsities, and contexts – from student education to engineering stress-strain tests. The utility of our algorithm was illustrated in two primary ways: first, the partitioning inherently provides insights into the dataset's structure. For instance, three distinct patterns were detected in the porous structure's stress-strain dataset: an initial hook, convex, and concave parts. Second, certain datasets can be learned more accurately with a modular model based on the active partitioning algorithm than with a single model. If a model's accuracy in learning a dataset is unsatisfactory and the dataset is likely structured along predominant patterns with little overlap, we recommend applying our pipeline of the active partitioning algorithm and modular model. Particularly in the context of expensive data points, improving the model on this path without adding more data points can be financially beneficial. In the future, we will explore a third application: adapting data collection strategies based on the active partitioning algorithm.

## 7 REPRODUCIBILITY

We have ensured that all presented results are easily reproducible. The partitioning algorithm is detailed with a flow chart (see Fig. 1) and pseudo-code (see Alg. 1) in Section 3. This section also describes the adding and dropping mechanisms, with their pseudo-code provided in the appendix in Section A.2 (see Alg. 2 and Alg. 3). The modular model is thoroughly explained in Section 4.1, including a flow chart (see Fig. 5). Table 2 in the appendix in Section A.2 lists all significant parameters used in the experiments with the partitioning algorithm and the single and modular model. All datasets used are properly cited in Section 4.2. The code is submitted as supplementary material.

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

# A APPENDIX

## A.1 ANALYSIS OF MODULAR MODEL PERFORMANCE

We observed that, for several datasets, the modular model utilizing the partitioning algorithm significantly outperformed the single model. To analyze these observations in more detail, we created the plots shown in Figure 8. We compared the performance of the modular and single models across ten test runs for each of the 25 datasets. The datasets with final losses displayed in the histograms in Figure 7 are marked with unique colors for identification, while all other datasets are illustrated in orange.

Figure 8: evaluation of the influence of multiple characteristics of the modular model on the performance of the modular model compared to the single model across all tested datasets.

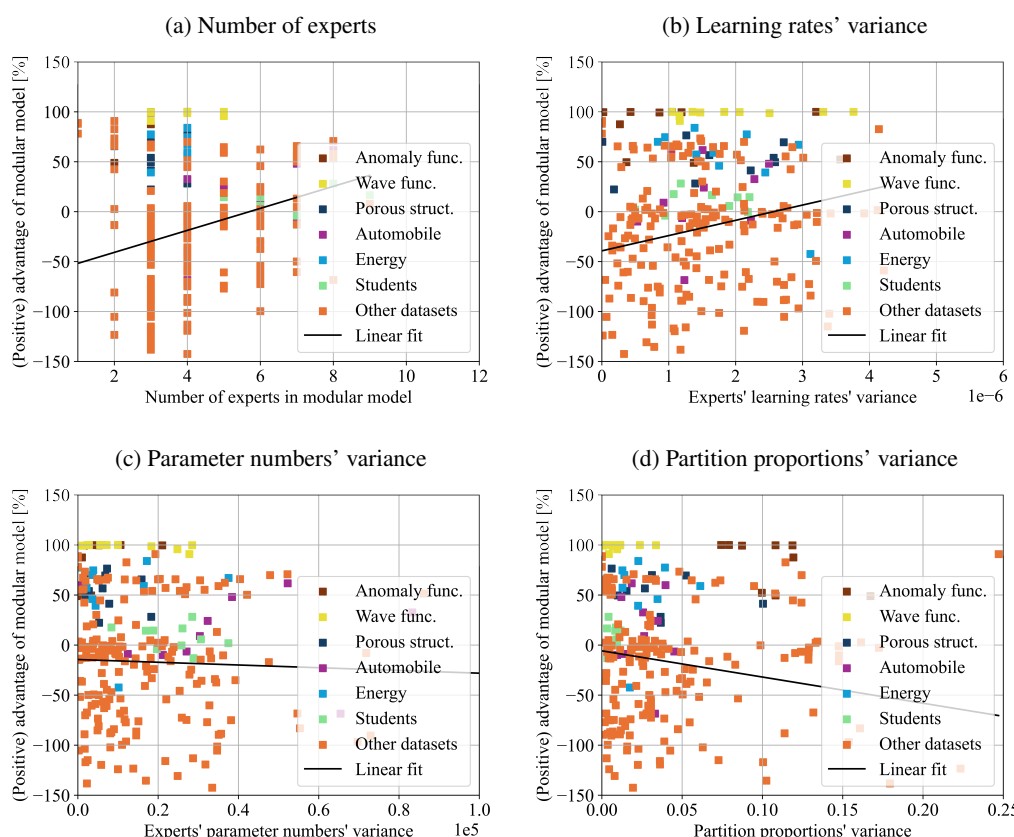

For each dataset, we computed the mean test loss of the single model over the ten test runs. We then compared the test losses of the modular model against this benchmark. For example, if the single models achieved an average loss of 100 and the modular model achieved a loss of 80, we recorded a performance value of 20%. Conversely, if the modular model achieved a loss of 120, we recorded a performance value of -20%. These performance measures were plotted against potentially influential parameters: the number of experts, the variance in the experts' learning rates, the variance in the experts' parameter counts, and the variance in the partition proportions.

Firstly, we observed that the performance of the modular model compared to the single model improves with an increasing number of experts (see Fig. 8a). Since the number of experts in the modular model corresponds to the number of patterns the partitioning algorithm has separated, this insight is more about the datasets that work well with this approach than about the modular model itself. The more separate patterns are found within one dataset, the better the modular model can be expected to work. Given our initial expectation that not all datasets would contain separable

patterns, this finding is not surprising. The more clearly a dataset is structured around separable patterns, the more effective our approach appears to be.

The modular model allows for the adjustment of hyperparameters locally for each expert, unlike the single model with a global uniform hyperparameter setting. In our experiments, we varied only the learning rate, the number of layers, and the number of neurons per layer. For this analysis, we combined the number of layers and the number of neurons per layer into a single metric: the number of trainable parameters. We evaluated the impact of locally adapting the hyperparameter settings to each pattern. The more the hyperparameter settings are tailored to each pattern, and the more they differ from a constant setting for the entire dataset, the greater their variance among all experts in a single run. Consequently, we plotted the performance of the modular model compared to the single model versus the variance in experts' learning rates (see Fig. 8b) and the variance in experts' trainable parameters (see Fig. 8c). We observed a moderate correlation between the modular model's performance and the adaptation of learning rates, but no correlation with the adaptation of trainable parameters. Notably, also with small variances in learning rates, modular models outperformed single models. We conclude that locally adapting learning rates to each pattern is moderately beneficial, whereas adjusting the number of layers and neurons per layer does not appear to have a significant impact.

Finally, we plotted the performance of the modular model compared to the single model against the variance in partition proportions for each run (see Fig. 8d). Our aim was to verify that the algorithm identifies significant patterns rather than just isolating small, difficult segments. Our findings confirm this hypothesis, indicating that the more uniform the partition proportions, the more effective the modular model becomes.

## A.2 Details on partitioning algorithm and modular model

For those interested in understanding the partitioning algorithm in full detail, this section provides the pseudo-code for the adding (see Alg. 2) and dropping (see Alg. 3) mechanism of the partitioning algorithm. Additionally, Table 2 lists all significant hyperparameter settings for both the partitioning algorithm and the modular model.

---

**Algorithm 2** Adding: train new model with badly predicted data points.

---

**procedure** ADDMODEL
    $allLosses \leftarrow$ Losses of best prediction for each data point
    $lossBound =$ mean$(allLosses) +$ std$(allLosses)$
    $dataPoints \leftarrow$ Data points with loss above $lossBound$
    $oldLoss \leftarrow$ Mean loss of $dataPoints$
    $newModel =$ new Model()
    $newModel$.train$(dataPoints)$
    $newLoss = newModel$.getLoss()
    **if** $newLoss < oldLoss$ **then**
        add$(newModel)$
    **end if**
**end procedure**

---

---

**Algorithm 3** Dropping: drop highly redundant models.

---

**procedure** DROPMODELS
    **for each** $dataPoint$ **do** $lossWithAllModels$ += lossOfBestModel
    **end for**
    **for each** $model$ **do**
        **for each** $dataPoint$ **do**
            **if** $model$ == bestModel **then**
                $lossWithoutModel$ += lossOfNextBestModel
            **else**
                $lossWithoutModel$ += lossOfBestModel
            **end if**
        **end for**
        $replacability = lossWithoutModel/lossWithAllModels$
        // 10% greater loss without model $\rightarrow replacability$ = 1.1
        **if** $replacability <$ droppingReplacability **then**
            drop($model$)
        **end if**
    **end for**
**end procedure**

---

Table 2: hyperparameter settings of the partitioning algorithm and the single and modular model during the experiments.

| | |
|---|---|
| Optimizer | Adam |
| Activation function | tanh |
| Epochs partitioning algorithm | 1,000 |
| Epochs modular model | 500 |
| Scaled feature range | [-1,1] |
| Batch size | 16 |
| Partitioning: initial model number | 10 |
| Partitioning: adding check | every epoch |
| Partitioning: dropping check | every epoch |
| Partitioning: dropping threshold | 1.8 |
| Hyperparameter search runs | 100 |
| Minimal layer number | 2 |
| Maximal layer number | 6 |
| Minimal neuron number per layer | 4 |
| Maximal neuron number per layer | 10 |
| Minimal learning rate | 0.0001 |
| Maximal learning rate | 0.005 |

