# OpenReview forum: "Active partitioning: inverting the paradigm of active learning"
_ICLR.cc/2025/Conference — ICLR 2025 Conference Withdrawn Submission_

### Official Review · Reviewer_8fUL · 2024-10-24

**Soundness:** 2
**Presentation:** 2
**Contribution:** 2
**Rating:** 3
**Confidence:** 4

**Summary:**

The paper introduces an algorithm that leverages competition between models to partition datasets based on distinct functional patterns. Unlike traditional active learning, which focuses on minimizing data for weak areas, this approach amplifies the strengths of models, promoting specialization. The modular models, consisting of multiple expert models each focused on learning a specific partition, demonstrate significant improvements over a single model.

**Strengths:**

1. The writing in this paper is easy to understand, and the use of flowcharts and other visuals makes it easier to grasp the core methods and concepts.

2. The authors provide pseudocode and detailed parameter settings in the paper, and the code is included in the supplementary materials, ensuring the reproducibility of the work.

**Weaknesses:**

1. The number of dataset partitioning baselines compared is insufficient. In the related work section, the authors discuss other dataset partitioning methods, while the authors did not compare active partitioning with any of these methods. The authors may supplement the baselines or explain why there is no comparison between them.

2. The modular model tends to underperform compared to a single model when the split dataset using active partitioning exhibits one coherent pattern or when multiple patterns have significant overlap. A related work [1] that first attempts to solve a problem with a single network and handles the unsolved portion(s) of the input domain recursively seems to be superior to the proposed method.

3. The authors claim that the novelty of this work lies in “the development of a flexible partitioning method through the competition of entire models,” but what advantages does this approach offer compared to previous dataset partitioning methods? The motivation behind the proposed method needs further elaboration and clarification.

4. The authors may provide a more detailed introduction to active learning and elaborate on how the proposed method invests in the paradigm of active learning.

5. The paper is lack of detail in the dataset partitioning phase. The authors mention that competing models might exhibit differences, such as using ‘wider neural networks or smaller learning rates’ for different patterns, but they do not provide concrete details on how model diversity is implemented during this stage. It would be beneficial to elaborate on how these variations are chosen and how they impact the effectiveness of the partitioning process.

6. The authors sometimes mix in-text and parenthetical citations throughout the related work section, such as "Wu et al.adapted ... (Wu et al., 2004)."

[1] V Scott Gordon and Jeb Crouson. Self-splitting modular neural network-domain partitioning at boundaries of trained regions. In: Proceedings of the 2008 IEEE International Joint Conference on Neural Networks, pp. 1085–1091, 2008.

**Questions:**

1. I would like to know how much time the active partitioning and training of modular model will cost compared to training a single model.

2. I wonder whether the competing models used for active partitioning can be directly used to combine a modular model.

---

> ### Author Response · Authors · 2024-11-18
>
> Thank you for taking the time to review our paper. We appreciate your constructive feedback. We will include additional baselines and further clarify and elaborate on the points you raised. Regarding your question about the computational cost, the modular model based on the partitioning result is significantly higher, although it has not been precisely measured. As for your question about whether the competing models could be used directly, that is indeed correct. Our aim was to clearly separate the partitioning algorithm from its utilization by a modular model to emphasize the generalizability of our partitioning result.

---

### Official Review · Reviewer_A2gz · 2024-10-27

**Soundness:** 2
**Presentation:** 2
**Contribution:** 2
**Rating:** 3
**Confidence:** 4

**Summary:**

This paper discusses a new learning paradigm called active partitioning, aiming to improve model performance by leveraging competition among models. The key idea is to separate and detect distinct functional patterns within datasets by rewarding models that provide the best predictions for specific data points with additional training on those points. This encourages each model to specialize in certain patterns, allowing the datasets to be divided into specialized partitions. Unlike traditional active learning, which focuses on training models based on their weaknesses to minimize training data, active partitioning emphasizes strengthening models' specialties. The approach is tested on datasets with distinct patterns (e.g., stress and strain data), showing how models can learn different partitions. The results demonstrate improved performance, with a 54% reduction in loss compared to single models handling the entire dataset, validating the effectiveness of active partitioning.

**Strengths:**

1. Interesting new paradigm. Even though it's similar to the ideas of mixture of experts which are well-studies in current LLMs era, the idea of applying multiple experts and partitioning datasets are interesting in active learning literatures.
2. The number of datasets in experiments section is impressive, including 2 two-dimensional datasets and 22 datasets from UCI Machine Learning Repository.

**Weaknesses:**

1. Lack of related works: The author mentions mixture of experts algorithm in Section 2.2. There is a rich body of related works regarding applications of mixtures of experts on LLMs [1, 2, 3].

2. Lack of theoretical justifications. Most of partitioning experiments have theoretical guarantees and more theoretical understandings would be helpful in understanding this algorithm.

3. Datasets are too simple and small scale. Code is not open-sourced. Datasets selected are mainly from UCI Machine Learning Repository where most of them are low-dimensional and small scale in terms of datasets size. Since there are no theoretical justifications, experiments should not be limited to regression tasks.

4. Ablation study of network architectures. The tasks should be not limited to regressions settings and more experiments regarding various network architectures should be discussed. The authors claim active partitioning paradigm is better than active learning but many active learning algorithms have experiments showcasing there optimality across multiple networks architectures. For instance [4] performs experiments across network architectures including networks similar to LeNet and ResNet-18.

References:
1. Gross, Sam, et al. Hard mixtures of experts for large scale weakly supervised vision. In Proceedings of the IEEE Conference on Computer Vision and Pattern Recognition, pages 6865–6873, 2017.
2. Zhou, Yanqi, et al. "Mixture-of-experts with expert choice routing." Advances in Neural Information Processing Systems 35 (2022): 7103-7114.
3. Riquelme, Carlos, et al. "Scaling vision with sparse mixture of experts." Advances in Neural Information Processing Systems 34 (2021): 8583-8595.
4. Ding, Zixin, et al. "Learning to Rank for Active Learning via Multi-Task Bilevel Optimization." The 40th Conference on Uncertainty in Artificial Intelligence.

**Questions:**

No

---

> ### Author Response · Authors · 2024-11-18
>
> Thank you for taking the time to review our paper and for providing valuable feedback and literature resources. We will incorporate datasets of larger size and higher dimension in future experiments. Additionally, we will investigate whether we can theoretically guarantee the convergence of our approach.

---

### Official Review · Reviewer_D6kP · 2024-11-03

**Soundness:** 2
**Presentation:** 3
**Contribution:** 1
**Rating:** 3
**Confidence:** 4

**Summary:**

The authors propose to partition the dataset by using predictions from multiple models.

During training, each sub-model is allowed to submit their predictions for all points in the datasets. The datapoints are then assigned to the sub-model with the best performance, and the sub-model is trained only these datapoints. As training proceeds, the hope is that the process induces specialization in the models, which is then translated into a partitioning. There is some connection to active learning, where datapoints are chosen for which the model is most uncertain about; whereas here, the datapoints are assigned to the model with best performance.

Experimental results are reported on 6 datasets, 3 of which are unidimensional datasets.

**Strengths:**

* The claims of the paper are easy to understand (though I dont quite believe them, see below)
* The experimental results one of the datasets was interesting to read

**Weaknesses:**

TLDR; I dont think the contributions of the paper meet the conference bar.

* There are lots of existing work on MOEs, this paper feels like re-inventing them from scratch. There is minimal mention to existing literature, no comparisons.

* The experimental results are quite unconvincing. The scale of the datasets are just too small. Why not have larger capacity models which can learn more. The scale of the datasets + model sizes (the latter I suspect is also small), makes me question if partitioning the dataset is needed at all.

* Even if we assume that partitioning is required, why not compare with simpler baselines like run clustering algorithm first, and then train independent models on the clusters?

**Questions:**

Please see weakness above.

---

> ### Author Response · Authors · 2024-11-18
>
> Thank you for taking the time to review our paper. We anticipated that a divide-and-conquer approach might be challenging for sparse datasets, such as the insurance risk dataset with 59 features and only 159 samples. However, our approach performed well even in these cases. We are confident that we can extend this performance to larger datasets and will include this in our future work. We also appreciate your suggestion for baseline comparisons and will incorporate them in our next steps.

---

### Author Response · Authors · 2024-11-18

Dear Reviewers, we would like to express our gratitude for taking the time to review our work. We truly appreciate your constructive feedback. We have decided to withdraw this version of the paper. We plan to address the points you raised in the next iteration.

---

### Note · Authors · 2024-11-18

**Comment:**

We truly appreciate the constructive feedback. We have decided to withdraw this version of the paper. We plan to address the points raised in the next iteration.

**Withdrawal Confirmation:**

I have read and agree with the venue's withdrawal policy on behalf of myself and my co-authors.